# National Level Support Programs for Youth in Relation to Effective School-To-Work Transition: Examples of Italy, Moldova, and Latvia

Maria Diacon [1,*], Liena Hačatrjana [2] , Victor Juc [1], Victoria Lisnic [1,*] and Antonella Rocca [3,*]

1   Legal, Political and Sociological Researches Institute, Moldova State University, MD-2009 Chisinau, Moldova; juc.victor@gmail.com
2   Faculty of Education, Psychology and Arts, University of Latvia, LV-1586 Riga, Latvia; liena.hacatrjana@lu.lv
3   Department of Management and Quantitative Studies, University of Naples Parthenope, 80132 Naples, Italy
*   Correspondence: mariadiacon@yahoo.com (M.D.); vikalisnic@gmail.com (V.L.); antonella.rocca@uniparthenope.it (A.R.)

**Abstract:** The school-to-work transition is one of the trickiest steps in the life cycle of an individual because when young people complete their education and enter the labor market, they have to compete to attain a job while lacking the job experience or skills required by employers. Across European countries, the school-to-work transition shows very different characteristics and durations, stemming from, among other factors, (1) the different provisions of services at the country level to help young people become oriented in the labor market, (2) different historical backgrounds, and even (3) the different capacities of education systems to provide the skills required by employers, despite the efforts to homogenize the national education systems, which started with the Bologna process. In this paper, we aim to compare various programs implemented within formal education at the macro-level in Italy, Moldova, and Latvia, three rather different countries in Europe, that have the goal of helping young people during various stages of this transition. The conclusion we can draw is that each of these countries needs to adopt a coordinated and integrated strategy of reforms aimed at (a) preventing early school drop-outs; (b) incentivizing the attainment of a university degree; (c) reforming school curricula; (d) closing the gap between education systems and labor market requirements; and (e) improving the services that help young people during the school-to-work transition.

**Keywords:** school-to-work transition; skills; education system

## 1. Introduction

For each country, the education system is the most important institution for the acquisition of an individual's human capital with a larger goal: to ensure sustainable economic growth. It represents the most determinant factor for each country's socio-economic development and helps to ensure a smooth transition from school to work [1,2].

At the European level, many efforts have been made to close the gaps within the different education systems. The Bologna process began in June 1999, and it tried to bring more coherence into higher education systems across Europe, but until now, many differences among them persisted [3]. These differences have been largely investigated by the current literature because they are connected with the capacity of young people to access the labor market during their school-to-work transition and with labor market characteristics.

In light of the drastic transformations that the labor markets have experienced in recent years, and the even more dramatic changes expected in the future, it is important that at the European level, some reforms are implemented in a coordinated way. This favors the convergence of the various education systems in a similar direction that will enable the

preparation of young people to satisfy the new labor market requirements in terms of skills and new paradigms related to overcoming the concept of permanent work.

This is of great importance, in particular, for the European Union countries, whose labor markets currently have no barriers, favoring the free circulation of goods, and even people, including young people during their job search activities. However, the analysis of the education systems and labor market characteristics, as well as the effort to overcome the still-existing significant differences, is even more important for countries who want to join the EU and, therefore, need to reach similar levels of competitiveness for enterprises and for people who are searching for employment. For the analysis of the characteristics of the different types of education systems in force in European countries, the socio-economic literature being referenced in this study starts from the pioneering studies of Esping-Andersen and includes many of his contributions [4–7], until the most recent study by Pastore et al. (2021), which refers more broadly to the concept of transition regime.

These transition regimes represent the outcome of the interactions of many factors influencing the labor market, the education system, and, more generally, the socio-economic framework in which people interact. However, these strong cross-country differences translate into the different capacities of young people entering the labor market when completing their studies and in the different outcomes in terms of educational levels attained [8,9]. To reduce the duration of the school-to-work transition and increase the efficacy of the education system, policymakers in each country are implementing many reforms and programs.

Having a cross-country perspective in the analysis of the existing reforms and measures adopted by governments to improve education systems and labor market functioning is very useful for identifying the most appropriate actions and best practices that could favor, even in other countries, an improvement in people's quality of life.

The aim of the current paper is to give insight into macro-level intervention programs aimed at youth in three countries, Italy, Moldova, and Latvia, in relation to the importance of an effective school-to-work transition for young people.

The choice of these countries is motivated by the need to compare interventions and data in countries with very different backgrounds: (1) Italy, one of the EU founders, showing the longest transitions from school to work in recent years, as well as the highest share of NEETs and one of the lowest shares of young people with tertiary education at the EU level; (2) Latvia, a country who regained independence after Soviet occupation in 1990, joined the EU in 2004, and has aspired to modernize their education system since then, also showing average or better levels for all these indicators at the EU level; and (3) Moldova, an EU candidate country, showing levels for these indicators is still far from the EU standards, which are in a state of constant improvement.

## 2. Theoretical Framework

European countries show very different characteristics in terms of the education system and labor market functioning. To explain and compare the different national frameworks, it is very useful to refer to the concept of transition regimes. Looking at European countries, we can identify the continental model of transition regimes, which includes the continental countries of Germany, Austria, and, more recently, France. It is based on a dual education system, with a high degree of overlap between work and school, and, consequently, a quick and successful transition from school to work.

The liberal countries of Ireland and the UK show instead a sequential education system of high quality, with few students starting to work during their studies, but a flexible and fast transition to work; the Mediterranean model connects to the so-called sub-protective transition regime. It is characterized by a rigid and sequential education system and a scarce connection of the educational contents with the labor market, in the sense that its contents appear strongly theoretical and very far from what the labor market requires.

Finally, Eastern countries show a single-structure education system, which in some countries is still derived from the legacy of the late communist system, or largely influenced

by centralized education management, while other countries are striving to modernize the curriculum and approach to learning (e.g., Latvia has recently started implementation of the competency-based education) [10].

The existence of these different transition regimes makes the goal of reducing the differences in the education systems of these countries not easy. In recent years, national governments have implemented different reforms to improve the functioning and efficacy of the education system and to reduce the existing differences with other countries. These reforms have been implemented at the micro, meso, or macro levels. In some cases, with reference to the EU countries, similar reforms have been suggested at the EU level and to be implemented in the same way in all the member countries. Some of them are intended for all the people living in these countries, while others are only for people living in specific regions or assessed as more in need, such as, for example, it is with the Youth Guarantee Scheme. The latter was introduced in 2014 in many EU countries looking at the successful experience of the Scandinavian countries who had already experimented with this reform in the previous years. Indeed, in some cases, especially for countries of medium and large size in terms of population and territorial extensions like in Italy, Germany, and Spain, the analysis of the education system and labor market functioning should be made at an intra-national level because many interventions are implemented for local and regional areas in view of the existence of strong differences in the education system, employment rates, and the types of intervention programs at this level. The reinforced Youth Guarantee, launched in 2020 in response to the crisis due to the COVID-19 pandemic, is an example of reforms addressed only to specific regions. Even the NEET rates concerning the share of young people not in employment, education, or training show strong intra-national variability [11,12]. A recent study by Errico et al. (2022) commissioned by the European Commission shows that Bulgaria and Italy are the countries with the higher regional variability in the NEET rates. The study describes the different ways each country, and in some cases even each region, had implemented the Youth Guarantee, with the help of the Public Employment Services.

Our scope is therefore to shed light on some criticalities of the education systems of the countries selected for the analysis and examine the programs and reforms adopted with the scope of improving them and therefore promoting the reduction of inequalities.

## 3. Empirical Context

The school-to-work transition can be defined as the period from the completion of the studies until the achievement of a stable job. It may be challenging because youths must compete to attain jobs, but they lack experience or the skills employers search for. Across European countries, the school-to-work transition shows different characteristics, based on different provisions of the labor market services, the capacity of education systems to provide the skills required by employers, and levels of unemployment [13]. A primary role in determining the different levels of difficulties met by young people to enter the labor market is played by the education system. While in some countries young people completing education are ready to overcome the barriers to entering the labor market and easily find a job, in other countries young people face difficulties because they need to acquire the skills and the additional competencies required by the labor market without any type of support [14]. It is important to analyze various indicators regarding this issue in different countries in relation to specific programs that have been active in these countries.

First, a brief comparison is given for Italy, Latvia, and Moldova, followed by examples of interventions from these countries. Table 1 shows several important indicators of the education system in the three European countries whose education systems are analyzed in this chapter: Italy, Moldova, and Latvia. The same indicators are reported with reference to the EU-27 area, which represents for EU countries the average value and EU partners as a sort of benchmark to close the gap with the EU. First, the amount of early school leavers is presented. As we can see from the table, the share of early school leavers in Italy is almost double the value registered in Latvia, while Moldova shows a value three times higher

than the Latvian one. Early school leaving is one of the main predictors of the NEET status because young people completing education only with the compulsory level of education meet strong difficulties in integrating into the labor market, whose complexity is increasing and will require even higher levels of skills and knowledge [15].

**Table 1.** Statistics data comparing Italy, Moldova, and Latvia in 2022.

| Indicators | EU-27 | Italy | Latvia | Moldova * |
|---|---|---|---|---|
| % Early school leaver (18–24 years) | 9.6 | 11.5 | 6.7 | 19 [1] |
| % Tertiary educated (30–34 years) | 42.8 | 27.4 | 45.7 | 30.1 [2] |
| Spending on education as % of GDP (2020) | 5.02 | 4.44 | 4.63 | 5.5 [2] |
| % Students underachieving in literacy (2018) | 22.5 [2] | 23.3 [2] | 22.4 [2] | 43 [2] |
| % NEETs (15–24 years) | 9.6 | 15.9 | 8.6 | 19.5 [3] |

* Data from Moldova are not in the Eurostat database. They are extracted from: Country Fiche 2020 Moldova-Education Training and Employment Developments.pdf (europa.eu, accessed on 16 August 2023). [1] The year of reference is 2017. [2] The year of reference is 2018. The share of students underachieving in literacy is extracted from Eurostat database from OECD PISA survey whose last available data are from 2018. [3] The year of reference is 2019.

Next, we can see in Table 1 that the share of people in tertiary education varies significantly among the three countries. While in Latvia one out of two young people have a tertiary degree, Italy and Moldova, which are 27.4% and 30.1%, respectively, show values considerably lower than the EU average of 42.8%. The differences in relation to the spending on education as a share of GDP are less marked while comparing the share of students underachieving in literacy; we can see that Moldova, with 43, shows a score that is more than double that registered in the other two countries, which are more in line with the EU-27 average of 22.5.

These few indicators make evident the strong difficulties of Moldovan and Italian education systems to retain young people in education for a longer period and, for Moldova, it shows even the existence of an important gap in the competencies acquired by the students. These criticalities are in strict relation with the very high rates of young people not in employment, education, and training (NEETs), which in Italy and Moldova are 19.5% and 15.9%, well above the corresponding values in Latvia and at the EU-27 level of around 9.6%.

## 4. Methodology

This research is focused on programs and policies developed in three countries: Italy, Latvia, and the Republic of Moldova. It is worth mentioning that two of these countries (Italy and Latvia) are European Union member states. A considerable quantity of data and information was taken from Eurostat and OECD databases. For the Republic of Moldova, a non-European Union member state we used the official statistics published by the National Bureau of Statistics. For comprehensive and in-depth research, the authors worked with analysis and studied the institutional websites of the resort institutions (ministries, agencies, and departments) and the literature on the characteristics of education systems.

After the separate analysis of the three countries, we tried to compare the reforms introduced and their effects. It is important to note that some reforms could have a great impact in some countries and a limited impact in others, depending on the general socio-economic framework in which they are applied. Another important consideration concerns even the different sizes of the countries analyzed.

Moreover, we followed the approach described by Lafton et al. (2023), who referred to the five steps to follow, as described by Seland et al. (2022) [16,17]:

1. Identification of keywords;
2. Using the keywords to search on the web within the documents, focusing on useful information for our goals;

3. Selecting studies, documents, and laws useful to describe the context in each country. To have a cross-country perspective, we selected studies and data from Eurostat and OECD;
4. Extract and merge the information useful to describe the education system and institutional framework.
5. Synthesize the main findings.

The research design included investigative techniques that allowed for a comprehensive view of the national programs for youth in relation to effective transition from school to work. The cooperation of the authors' team contributes to the identification of state capacities at the national level to plan, implement, and validate strategies that are necessary for a more educated and more developed society, from economic, social, and demographic points of view.

## 5. Countries' Analysis of the Education System, Recent Reforms and State-Wide Programs

In this section, we will present and then compare various programs activated in Italy, Latvia, and Moldova to improve the education system characteristics with the objective of tackling the risks for young people and facilitating the entrance of young people into the labor market.

### 5.1. Case of Italy

Across European countries, Italy shows one of the longest school-to-work transitions [8,14]. The causes of this negative primate are manifold and well-known in socioeconomic literature. At the macro level, they are mainly ascribable to the labor market characteristics and the education system. The Italian labor market shows high levels of unemployment and under-developed institutions regulating the school-to-work transition [18]. For the education system, we can say that it is of high quality, but very selective, because few young people reach a high level of education. Further, it is strongly disconnected from the labor market, in the sense that it is unable to transmit to young people the skills and competencies required by employers [19].

To fight against these recognized weaknesses of the Italian education system, in recent years, different governments have contributed to introducing many reforms. They have been mainly oriented to increase the degree of school autonomy and the opportunities for professional training and apprenticeship provided during the curricular path [20].

Among these reforms, "Alternanza scuola lavoro" (ASL) was introduced by Law 107/15 paragraphs 33–42 in 2015. More recently, in 2018 the law n. 145 of 30 December modified the ASL introducing the PCTO, "Percorsi per le competenze trasversali e l'orientamento" (in English: "Paths for Transversal Skills and Orientation"). These reforms are finalized more specifically to transmit transversal competencies and provide a guide to young people for their future careers.

Depending on the type of school orientation (general or technical path of studies), they (ASL first and PCTO after) consist of the introduction of further hour courses of compulsory nature in the curricula of high school students, to be carried out at least partially in working contexts. The main objective is therefore to offer students the possibility to spend additional time, besides the usual time spent at school, in practical activities, usually in an enterprise or institution connected to the school's specific field of study [21].

This reform has interested only students from schools whose duration is five years but limited to the last three years. To better understand the rationale of this reform, it is important to understand the characteristics of the Italian education system. It offers three different types of high schools:

- Lyceums, mainly characterized by theoretical contents, finalized to prepare students for university courses;

- Technical institutes, including both theoretical and technical subjects but specialized in a given field, such as trade, informatics, and so on. After their completion, it is possible to attend university;
- Professional institutes, finalized to prepare students for work, including very practical subjects. They allow students to attend university after their completion only if their duration is five years.

The number of additional hours provided with ASL ranged from a maximum of 400 for the high schools with professional or technical content to a minimum of 200 to be distributed in the lyceums. The ASL reform of 2018 that introduced the PCTO, further increased the differentiation among the different types of high schools in the number of hours, but reduced the number of hours as follows:

- A total of 210 h in professional institutes;
- A total of 150 h in technical institutes;
- A total of 90 h in the lyceums.

For the lyceums, the application of PCTO has been translated in many cases into activities made in collaboration with universities. Therefore, this action precisely connects with the orientation for the type of university to choose. Conversely, especially for the professional institutes, these activities have been translated in many cases into work experience in the factories.

Indeed, the primary scope of PCTO is to make the education system closer to the business, prepare students for work, increase innovation, and enhance the role of all the players concerning their functions and skills. One of the innovative inspiration criteria was to connect the knowledge and competencies that students learn at school with the applicative aspects of the work environment. This action has been pursued through the following specific goals:

- Make young people closer to the culture of work through stages and apprenticeship experience;
- Enrich the training offer, make it even more productive and near to the student's expectations and the needs of the socio-economic context;
- Make the learning process more flexible;
- Connect the experience with local development;
- Make the work environment even more open to the learning processes.

To give the activities undertaken during the PCTO more importance, they have been included in the high school Final Exam by the Delegation Law 384/17.

However, to be efficacious, this reform still needs substantial improvements. Indeed, as it is, it requires that each high school enter into an agreement with local entrepreneurs and other types of institutions to provide these additional activities. To make this possible, teachers from schools, who assume the role of internal tutors, should collaborate with teachers and experts from the external environment, who assume the role of external tutors, to define the best programs and training to offer students. However, these activities have not been effectively regulated. Further, in many cases, the lack of a sufficient number of enterprises and other institutions available on the territory makes the realization of these types of activity very difficult [22].

From the teachers' perspective, it has been translated into an additional workload even in terms of responsibilities. Further, the internal and external tutors identified as organizers for all the activities should have specific high competencies that they usually do not have, and no training has been planned until now for them [23].

For the students, especially those from technical and professional institutes, these experiences are associated with the same risks connected with regular work activities. In February of 2022, during the completion of a PCTO activity within a factory, there was a mortal accident involving a student. This fact aroused many protests from both students and teachers, which identified many criticisms on how ASL first and PCTO after were implemented. First, schools have the task of choosing the activities that students should

do during the ASL, and they, alone, are often unable to identify the best solution for them. Second, if the intention was to make the Italian education system more like the German one, a relevant difference between the Italian education system compared to the German one consists of the fact that in Germany, young people start to work effectively during school attendance. They are paid for this, while in Italy this option has not been considered at all. Most students, especially those from the lyceum, have assessed this experience as a useless waste of time [24].

It is, therefore, evident that a lot of work is still needed to ensure that the experiences of young people during the PCTO will be useful to acquire those transversal competencies able to enrich their practical skills, rather than translating into labor at no cost, to be used in the most disparate and useless activities for the training of students, if not those that are downright dangerous.

*5.2. Case of Moldova*

In the Republic of Moldova, as in other European countries, it is recognized that youth employment is a precondition for poverty eradication and sustainable development. Identifying the nature and extent of youth employment issues at the national level is extremely necessary for the formulation of integrated policies and intervention programs. Improving the school-to-work transition is a precondition for helping young people overcome difficulties in finding and maintaining a decent job.

The methodology for researching the transition from school to work has been developed by the International Labor Office and it enhances research on the situation. Young people aged 15–29 in the process of school-to-work-transition can be divided into several groups (data taken from the statistical survey "Transition from School to Work (TSM)") [25]:

- A total of 29.9% have completed the process, already having a satisfactory job (average age 25 years);
- A total of 26.6% are still in transition: (a) looking for a job; (b) having a job not satisfied with; or (c) not working or learning, but intending to work (average age 23.5 years);
- A total of 43.5% have not yet entered the transition process because (a) they are in training or studies, or (b) they neither study nor intend to look for work (average age 21.9).

The situation of young people in the labor market in the Republic of Moldova can be largely characterized by one term: discouragement. Young people are discouraged by the conditions of the labor market both in terms of taking the job and staying in the same position. According to the data of the Transition from School to Work study (NBS, 2015), depending on the reason for refusal, 76.5% of young people refused a job due to low salaries. The same phenomenon of discouragement can be observed among young people even in the school period when they realize that the training they obtain in educational institutions does not correspond to the requirements of the market. Thus, about 19.9% of young people who leave school early do so for economic reasons, 14.9% do so because they are not interested in studies, and 10.7% do so because they want to start working. Paradoxically, the same study shows us that about 29.1% of young employees are overqualified for the positions they occupy and only 2.1% are undereducated for the job they occupy.

According to this study in terms of the transition of young women and men to the labor market in the Republic of Moldova, only one out of two young people from the Republic of Moldova is employed based on the field of study. They are also discouraged from taking up a formal job, where the payment of taxes and low salaries push them to plead for an informal job. Another aspect of the large number of employed young people who have an informal workplace is the structure of the Moldovan economy, with a high proportion of seasonal agriculture, which forces a large group of young people, especially from rural areas, to exercise seasonal activities in agriculture [26].

Additionally, young entrepreneurs are unmotivated to start businesses, because of unattractive business conditions, limited economic opportunities, and low incomes. The most significant challenges of young people in business development are in proportion.

That is, 34.3—market competition, where young people have to compete with entrepreneurs well established on the market; 34.0—insufficient financial resources; 8.4—political uncertainties; and 7.2—insufficient expertise, but also legal regulations (1.6%).

Young people (16–29 years old) leave their first job motivated by the fact that they are not satisfied by their salaries (22.9%), pregnancy/birth of a child (17.3%), a better job (16%), or even go abroad to work or look for work (14.6%), and 53.3% of young employees who would like to change their job would do so for a higher salary [27].

Young people are not motivated to stay in the country to work, pleading for a well-paid job abroad, which would allow them to create a family or support the existing one. Migration is not only international in nature, young people also leave from the village to the city, within which there are several factors such as the lack of jobs and low wages, but also the lack or poor efficiency of social services (hospitals, schools, pharmacies, kindergartens, etc.). A good part of them live on the basis of remittances from family members working abroad.

Youth policy is an intersectoral field and, respectively, requires the cooperation of several state institutions for the development and implementation of youth activities. National legislation defines youth policies as "a set of principles, methods, and measures aimed at ensuring young people opportunities for participation, well-being, personal and professional development".

The National Youth Sector Development Strategy for 2014–2020 (SNDST 2020) is a document that includes the state policies to ensure the direct involvement of young people, youth organizations, and other actors of immediate tangent in the political, economic, social, and cultural life of the country [28].

The strategy provides for four priority directions of intervention, namely:

- Promoting the participation of young people in decision-making processes;
- Diversification and consolidation of services for young people;
- Development of economic opportunities for young people;
- Consolidation of the youth sector (SNDST, 2020).

First is the National Employment Strategy for 2017–2021 (SNOFOM). The National Youth Sector Development Strategy for 2014–2020 was one of the premises of a more in-depth evaluation and the establishment of young people as one of the main beneficiaries of the National Employment Strategy 2017–2021. Young people are well reflected in the strategy, namely in the major targets of the strategy, which proposed to increase the youth employment rate by 4.2% and reduce the youth unemployment rate to 7% (young people aged 15–29). SNOFM is the first public policy document that addressed young people in the NEET category, and, even more, they established specific indicators for them: NEET men, NEET women, rural, and urban [29].

Next is Law No. 215 of 29 July 2016, regarding youth. This law aims to establish the priority directions for the promotion and implementation of state policy in the field of youth in accordance with the current interests and needs of young people by creating a framework and providing a clear mechanism for their support and promotion. Additionally, the law provides for stimulating the entrepreneurial initiatives of young people, facilitating their employment in the labor field, capitalizing on skills, creating an appropriate environment for physical development, and promoting a healthy lifestyle [30].

Third is Government Decision No. 664 of 3 June 2008 regarding the National Youth Economic Empowerment Program (PNAET). The PNAET program is the only program aimed directly at young people through which, until 2016, 1678 projects of young beneficiaries were financed in a total amount of MDL 482.2 million [31].

Next is Law No. 179 of 21 July 2016 regarding small and medium enterprises (SMEs). This law establishes that young people are some of the main beneficiaries of SME development programs, in addition to returned migrant workers [32].

Finally, the National Regional Development Strategy (NRDS). The most seriously affected are the youth from rural areas, who have the fewest employment opportunities. Creating the conditions for the sustainable development of the country's rural areas must

be the cornerstone of any youth employment policy. NRDS has such an objective, that the NRDS action plan is focused on solving specific problems with available public resources and not on developing new mechanisms [33].

In the context of a general approach to young people, and especially in the context of a lack of norms and provisions that would protect and promote young people at the level of collective labor contracts and collective agreements, it is very difficult to identify well-targeted programs for the employment of young people in the labor field. However, some youth entrepreneurship programs have been identified and some youths are among the main beneficiaries.

The business incubator is an organization designed to support both the successful establishment of businesses and their future development. Most of the time, the incubator provides the physical infrastructure necessary to run the business, business consulting services adapted to the service of each company, and opportunities for establishing contacts (networking).

The incubators are of several types, varying, in particular, depending on the scope of the incubation programs they offer, their internal organization, the economic sector in which they are specialized, and the type of clients to whom their services are offered. Incubated companies are housed in the incubator for a period of 3 years. In this interval, companies benefit from services that can be grouped as follows:

- Offering space for rent for offices and production, at a price lower than the commercial price;
- Administrative and technical services;
- Consulting and guidance in business;
- Access to financing.

According to ODIMM reports, within business incubators by 2021, 11 business incubators hosted 268 companies, of which 144 were owned by young people [34].

Online employment platforms are a very effective tool for facilitating contact between employees and employers. These platforms are user-friendly (adapted for mobile devices) and convenient, which makes it even easier to match demand with supply in the labor market. Given the fact that each vacancy has specified requirements and rigors that must be met, it is very easy for those looking for a job to apply for a position that corresponds to their level of training. For employers it is just as simple—they put out a single ad and received hundreds of applications—especially since in the Republic of Moldova the number of jobs is clearly lower than the number of unemployed.

In the Republic of Moldova, there are many employment platforms in the field of work that are newer and aimed at young people. Young people, being the closest to information technologies, should have been the main beneficiaries of these platforms. In reality, however, the situation is a little different, with employers being very meticulous in setting the ideal profile of young people. With many options, they often hire experienced adults or even young people with higher qualifications than the job would require. That is why the efficiency of these platforms, in the context of employing young people in the labor market, is relatively low. The online environment counts over 20 platforms (www.jobs.diez.md, www.civic.md, www.jobinfo.md, accessed on 28 May 2022).

Career guidance centers. Within the project "Reconceptualization of Professional Guidance and Career Counselling" (REVOCC), three career guidance centers were created within three territorial agencies of ANOFM (Chisinau, Soroca, and Cahul). The project aims to study professional orientation and career counseling in order to identify solutions to existing problems, based both on the reality in the Republic of Moldova and on international experience. Beneficiaries of the Centers are people looking for a job, students, and young people seeking to explore opportunities in the labor market, the diversity of professions/trades, the assessment of their own skills and professional abilities, as well as professional guidance in order to achieve successful socio-economic integration [35].

Young people disadvantaged in the labor market (without previous experience, without education, from rural areas, etc.) need to be supported. They are part of society and can effectively contribute to the development of the environment they come from. Their

employment in the field of work will allow for an increase in revenues to the state budget through the payment of fees and taxes and will decrease the massive emigration of the population abroad.

*5.3. Case of Latvia*

There are various approaches implemented in European countries, Latvia among them, to tackle and prevent the problems with young people's transition from school to work. As mentioned in the introduction, one of the important first steps in this process is the formal education system and its capability to provide students with the necessary skills to be ready for the "real world" [20]. With this aim, a competency-based approach was introduced to Latvia's education system by changing the whole curriculum and the approach to studying, including the need to develop transversal skills [10].

However, the improved curriculum was just recently adopted [36,37], thus raising the issue of the quality of skills of those people who have already finished school. Therefore, the next important step is, of course, the implementation of various programs that help young people find jobs, develop their small businesses, or gain professional skills. For example, the "Youth Guarantee" project implemented by the State Employment Agency of Latvia reached about 20 thousand young persons in the period of 2014–2018 (nva.gov.lv/lv/node/246, accessed on 1 September 2023).

Another important part of achieving better outcomes regarding the issue is to work on the prevention of the problem of students' early school leaving. Prevention of early school-leaving is important, as leaving school leads to social and economic consequences, by limiting students' further options for education and work, thus generally increasing inequality risks [38].

To tackle the issue of early school leaving, an important intervention program has been implemented since 2017 in Latvia that aims to target even the youngest parts of the population. Support program "Pumpurs" in Latvia ("Bud" in English), funded by the European Social Fund (project Nr. 8.3.4.0/16/I/001), is mainly aimed at reducing early school leaving in all stages of secondary education; thus, the students that are currently still studying are involved. The period of the program is 2017–2023; therefore, we can assume that the first results already might be reflected in national-level statistics. The program is aimed to help at the stage when school-leaving can be prevented; thus, students do not cut off their further possibilities to study, obtain higher education, and later gain better-paid jobs [39].

It can be seen in Table 1 that in Latvia the number of NEETs is relatively lower than in the other countries. However, statistics show that the number of school leavers is higher in rural areas, at 6.2% in cities and 13.4% in rural areas, imposing a negative impact on their future [40,41]. Therefore, prevention is especially important in rural areas. This means that real results can be achieved if all parts of the country are equally involved. This was actually the aim of the project developers.

It was identified by developers of the project "Pumpurs" that working with several parties and stakeholders, including students and teachers, is necessary to tackle the problem, as it is ineffective to work in a fragmented manner. The project promotes the development of a system of sustainable cooperation between the municipalities in all of Latvia, educational institutions, teachers, support staff, and parents (or representatives of the students) to identify students at risk of dropping out in a timely manner and provide them with personalized support [39].

The work with prevention of early school leaving and the consequent problems can be effectively implemented when a strong collaboration with schools and other stakeholders is ensured. "Pumpurs" was developed as a national-level program, and any school and local municipality can become a project partner aiming for all regions of Latvia to be covered, including the rural areas. During the project activities teachers and professionals, as well as the target students can receive various consultations and support, including financial support for the target student to participate in the activities, training, and other support

options that are defined as necessary for the particular student. For each student that is participating a personal development plan is prepared; therefore, the help meets the individual needs of a student. These personal plans for interventions are implemented and supervised by school representatives. In 2020/2021, a total of 19,757 individual assistance and development plans were prepared as a part of the project [39].

An important feature of this prevention project is the identification of students with difficulties. Early detection of students at risk and prevention activities are crucial for the future of these students, so it is important that schools and teachers can receive systematic and high-quality support that is well-planned and targeted (also including psychological and educational support regarding work with problematic students). As a part of the program, teachers are also receiving consultations and supervision about the best methods to use to work with the students and their problems. In 2022, more than 1000 supervisions had already been carried out in 567 educational institutions in Latvia. This is equally important as helping the students, as it may reduce risks for the burnout of those teachers who are very involved and doing their best to help students [39].

However, one of the issues of projects like this is their continuation and sustainability, as a large-scale network has been developed and resources invested in implementing the effective system for working with youth; however, the program has its funding period that eventually comes to an end. It is crucial that programs like "Pumpurs" continue to operate after their current funding period finishes, in one form or another. State support is very important both financially and politically in this case (by defining youth issues as particularly important for the future). If we consider inequalities in the options for extra training and support that, for example, parents can offer for their children during the school years or after they finish school, it is important that such support programs take place. In the midterm evaluation (from The State Education Quality Service), the need for successfully continuing the cooperation established during the project was stressed [42]. Programs like these are and will be a great aid to teachers and schools in helping the students at risk, as for some students the school is the only option for the so-called "social lift" to achieve more in their lives and to overcome obstacles they might face.

In Latvia, there are several large-scale projects that have a focus on youth topics and issues, both in the prevention stage of school leaving and supporting the development of skills of young people and in the stage of actively tackling already existing problems with NEETs and transitioning to work. It can be seen that project "Pumpurs" in Latvia has been among the most important projects in preventing risks of dropping out of school during recent years [39], based on the principle of close local collaboration between municipalities and schools, at the same time with centralized supervision by a state authority. Based on the comparison between the countries reviewed in this chapter, it might be said that such prevention projects are effective.

### 5.4. Discussion about the Three Presented Examples of Italy, Moldova, and Latvia

The background, recent reforms, and current projects relevant to youth in their transition to become economically active members of society were presented in the previous parts of the paper. Vast differences were observed regarding tertiary education attainment, early school leaving, and other indicators (Table 1). The historical and cultural context cannot be ignored when comparing these countries (described in more detail in the introduction) in relation to the education and employment of young people.

One common feature in all countries strive for some kind of change in education to make it more modern, real-life related, and relevant for future needs. Education can thus be defined as a crucial and central part of helping young people from the early years of their lives until they get university degrees or develop their skills as adults. In the case of Latvia, it is a gradual change to competency-based education, also focusing on transversal skills. In Italy, it is a strive to make the education process more relevant and connected to real work settings. In the case of the Republic of Moldova, education is seen as an instrument of economic growth, the more educated the society, the higher the rates of

economic development will be. As all three countries are a part of Europe, thus reflecting democratic and modern values, it can be noticed that similar features regarding the goals of education are showing in all of them—however, with slightly different focus and timing.

The second common feature in the three countries is that the issues of school-to-work transitions are recognized and in general there is an aim to tackle them, through various state-wide programs and projects. Both in Moldova and Latvia, there are some forms of business incubators or similar supportive programs for young people who want to start their businesses. However, again we can conclude that the historical, cultural, and economic development factors cannot be ignored, as we see from data that, for example, in the Republic of Moldova an important reason for the demotivation of young people to stay in employment or to develop their own businesses, is the low level of salaries, and a proportion of them also tend to leave the country for a better-paid job. In contrast, in Latvia, a major wave of such migration was noticed after the economic crisis in 2009. Meanwhile, in Italy, the problem for youth is more with the long period of transitioning from school to work, and thus also becoming active participants in the economy. Therefore, we see that the developmental situation of each country is affecting the processes with youth, their attitudes, and their motivation for work.

To conclude, we can see that focusing on (1) the education system and content, and (2) focusing on helping youth integrate into the labor market, are the main means for supporting effective school-to-work transition in the three compared countries in Europe. The current study contributes to the identification and discussion of a wide variety of capacities at the national level to plan, implement, and validate strategies that are necessary for a society to become more educated and more developed, from economic, social, and demographic points of view.

## 6. Limits and Future Research

The analysis realized in this article is based on the description of the context and policies in force in each of the countries analyzed: Italy, Latvia, and Moldova. Despite the significant differences among these countries, arising from the different socio-cultural contexts, some common traits can be identified among them, and some characteristics observed in the performers' countries could act as an example of best practices for the others. However, additional future research must be implemented to measure the impact of the described policies in each country. To make this possible, we need specific data at the micro and macro levels for each country and a longer period of observation.

## 7. Conclusions

In recent years, the European labor markets have been subject to many transformations, and further dramatic changes are expected in the future. One of these deep transformations concerns overcoming the concept of a permanent job. These transformations require significant changes in the education system that should adapt to the changing world. Various countries around the world have already started to implement changes in their education systems to provide students with a meaningful set of skills and knowledge.

At the EU level, in the future, the process of transformation and homogenization of the state members' education systems, starting with the Bologna process, will produce further radical changes in the education and training systems. The main goal of these reforms is to spread the culture of the business and the market in a capillary way and train young generations capable of adapting to a market of increasingly complex work. The latter requires adapting to increasingly flexible working conditions, even to very different situations over time. Further, a similar scenario will demand from young people very high capabilities of adaptation in terms of availability to move even far from the country of origin or to work remotely or in hybrid forms.

In this paper, we analyzed and compared some recent reforms in the education system at the macro level in three European countries: Italy, Latvia, and Moldova, whose main characteristics and differences were explained in the introduction part. Major differences

can be seen in the three countries (Table 1) and there are various interventions in these countries that can be defined by different goals: (1) aim at helping young people find jobs or train specific skills for the job market; (2) focus on the prevention of early school leaving to reduce further risks.

For example, in the case of Latvia, the prevention program for early school leaving might have been helpful, as the rates of school dropouts are lower compared to other countries, further leading to a higher percentage of students in tertiary education that would not have been possible if they had dropped out of high school. For Italy, the idea to make the school system less disconnected from the labor market could reduce the difficulties for young people, but it must be accompanied by other reforms to fight school leaving in the first place. Regarding Moldova, various challenges were identified with involving young people in the labor system, and there are many programs starting to be implemented that are aimed at tackling these problems.

However, when reforming the education system, policymakers should take in mind and protect the primary role of education: providing the skills required by the labor market, adapting to the flexibility, and structural and technological innovations are important goals, but above all the education system should prepare students to fulfill the real needs of the society, transmitting the ethical principles beyond the business interests. In this direction, the reforms finalized to reduce the share of early school leavers are very important because highly educated people contribute to improving society's functioning and ideals.

To conclude, it is important to work on both the prevention and the consequences of youth school-to-work transition issues, as results can be achieved at the state level if proper work is invested.

**Author Contributions:** Conceptualization, A.R. and M.D.; methodology, A.R., M.D. and L.H.; writing—original draft preparation, A.R., M.D., L.H., V.J. and V.L.; writing—review and editing, A.R., M.D., L.H., V.J. and V.L. All authors have read and agreed to the published version of the manuscript.

**Funding:** This publication is based upon work from COST Action CA18213 Rural NEET Youth Network, supported by COST (European Cooperation in Science and Technology).

**Institutional Review Board Statement:** Not applicable.

**Informed Consent Statement:** Not applicable.

**Data Availability Statement:** No new data were generated during this study.

**Conflicts of Interest:** The authors declare no conflict of interest.

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
