# Peer review of "National Level Support Programs for Youth in Relation to Effective School-to-Work Transition: Examples of Italy, Moldova, and Latvia"

_societies, doi:10.3390/soc13090208_

Round 1
Reviewer 1 Report
Dear Author,
I hope this message finds you well. I have had the opportunity to review the article that has been assigned to me for evaluation. However, after a detailed analysis, I have some fundamental concerns that lead me to the decision of rejecting the article in its current state.
One of the main reasons for my rejection is the absence of a methodology section in the article. Methodology is crucial in any academic study as it provides a solid foundation for the analysis and interpretation of the results. Without a dedicated methodology section, the validity and reliability of the presented findings cannot be determined. Therefore, I consider it essential to include a methodology section in the article to ensure its academic rigor.
Furthermore, throughout the article, there are assertions that are not adequately substantiated. The lack of evidence and reference to reliable sources undermines the credibility of the presented arguments. For the article to be valid and convincing, it is crucial to support all claims with previous research, empirical data, or other relevant academic resources. Without this solid foundation, the proposed conclusions and recommendations lack support and may raise doubts among readers.
Additionally, it would be beneficial to delve deeper into the analysis of the causes behind the differences in the school-to-work transition among the mentioned European countries. Merely mentioning the existence of different services and educational capacities does not provide a comprehensive understanding of the underlying factors. It would be advisable to investigate and discuss in greater detail government policies, specific educational programs, and other relevant variables that influence these differences. This would allow for a more accurate and comprehensive assessment of the situation.
In summary, based on the lack of a methodology section, unsubstantiated assertions, and a lack of thorough analysis of the causes, I have decided to reject the article in its current form. However, I encourage you to review and address these concerns before resubmitting the article for consideration. With appropriate improvements, the study can become a valuable contribution to the understanding of the school-to-work transition in different European countries.
Thank you for your understanding, and I wish you success in future revisions and publications.
Sincerely,
Reviewer
Author Response
Dear Author,
I hope this message finds you well. I have had the opportunity to review the article that has been assigned to me for evaluation. However, after a detailed analysis, I have some fundamental concerns that lead me to the decision of rejecting the article in its current state.
One of the main reasons for my rejection is the absence of a methodology section in the article. Methodology is crucial in any academic study as it provides a solid foundation for the analysis and interpretation of the results. Without a dedicated methodology section, the validity and reliability of the presented findings cannot be determined. Therefore, I consider it essential to include a methodology section in the article to ensure its academic rigor.
Thanks for this observation. The methodology section showing the approach of analysis used in the article has been developed as requested.
Furthermore, throughout the article, there are assertions that are not adequately substantiated. The lack of evidence and reference to reliable sources undermines the credibility of the presented arguments. For the article to be valid and convincing, it is crucial to support all claims with previous research, empirical data, or other relevant academic resources. Without this solid foundation, the proposed conclusions and recommendations lack support and may raise doubts among readers.
According to this suggestion, we revised the whole article and for each affirmation we indicated previous research supporting our claims. The new references have been highlighted in red to be immediately evident
Additionally, it would be beneficial to delve deeper into the analysis of the causes behind the differences in the school-to-work transition among the mentioned European countries. Merely mentioning the existence of different services and educational capacities does not provide a comprehensive understanding of the underlying factors. It would be advisable to investigate and discuss in greater detail government policies, specific educational programs, and other relevant variables that influence these differences. This would allow for a more accurate and comprehensive assessment of the situation.
The discussion about the national and regional differences in the education system and labour market characteristics has been deeper developed as required. We provided indeed a new section 2 with an appropriate theoretical framework, starting from the description of the different types of transition regimes and then discussed on the type and possible effects of the reforms that could be implemented at the regional or national level. Even the introduction has been modified describing more in detail the efforts to reduce the gaps between the education systems, starting from the Bologna process opened in 1999.
In summary, based on the lack of a methodology section, unsubstantiated assertions, and a lack of thorough analysis of the causes, I have decided to reject the article in its current form. However, I encourage you to review and address these concerns before resubmitting the article for consideration. With appropriate improvements, the study can become a valuable contribution to the understanding of the school-to-work transition in different European countries.
Thank you for your understanding, and I wish you success in future revisions and publications.
The authors thank the reviewer for these valuable suggestions.
Reviewer 2 Report
1. I think that the first paragraph of the introduction needs a couple of references about the importance of education system. The authors could use the following:
Alegre, M., Casado, D., Sanz, J., and Todeschini, F. (2015) The Impact of Training-Intensive Labour Market Policies on Labour and Educational Prospects of NEETs: Evidence from Catalonia (Spain). Educational Research 57: 151–167.
Farrugia, D. (2018) Spaces of youth work, citizenship and culture in a global context: Youth, young adulthood and society. Routledge.
2. I liked the international comparison of educational systems and youth transitions across European countries. I suggest the authors also refer to intra-national variation of educational systems, youth employment patterns, intervention programs, youth transitions and youth inactivity across the European regions. Literature on this issues has been recently increasing and highlighting this would place the paper better in the wider academic debates, thus increasing its value and contribution. Illustrative works that could be used include the following:
Avagianou, A., Kapitsinis, N., Papageorgiou, I., Strand, A.H., and Gialis, S. (2022) Being NEET in Youthspaces in the EU South. A Post-recession regional perspective. YOUNG 30, 425-454.
Kapitsinis, N., Poulimas, M., Emmanouil, E. and Gialis, S. (2022) Spatialities of being NEET in an era of turbulence: a regional resilience perspective for the Mediterranean EU South. Journal of Youth Studies https://doi.org/10.1080/13676261.2022.2101355
3. The section ‘Theoretical Framework’ should be renamed. It does not present theories or concepts, rather than the empirical background of the case study countries vis-à-vis the European context. Therefore, I suggest renaming it to ‘Empirical context’
4. The main weakness of the paper is the lack of a conceptual framework (which is a necessary ingredient for an academic paper) introducing the main concepts and theories about the education system and school to work transition. The authors should write a section between the introduction and the current second section which is called ‘Theoretical Framework’ but does not present theories or concepts, apart from the first paragraph.
5. The ideas and arguments of the conceptual framework should be linked to the findings in the section of the analysis, by adding an extra section between the analysis and conclusion that will discuss the results in depth, connect them with the conceptual framework and then highlight their importance for the wider academic debates on education systems and their international variation. In other words, the paper lacks engagement with relevant literature (analysis, discussion and conclusion). This should be strongly revised and significantly expand the analytical character of the manuscript, which at the moment is rather descriptive.
6. Additionally, the authors should add a brief section where they present and explain the methodological steps they followed to conduct this research and write this paper.
7. Finally, a basic issue that the authors should provide their own explanation in the conclusion, based on the results, from a critical point of view is what the national government should do about the relationship between education and labour market: should the education system be subordinated to the labour market needs, should the education system be more connected with the labour market needs or should the education system prepare students to fulfil the real needs of the society beyond the business interests? I think this question about education system vis-à-vis labour market is key when presenting and explaining studies related to contemporary education.
Some minor editing of English language would improve the paper
Author Response
I think that the first paragraph of the introduction needs a couple of references about the importance of education system. The authors could use the following:
Alegre, M., Casado, D., Sanz, J., and Todeschini, F. (2015) The Impact of Training-Intensive Labour Market Policies on Labour and Educational Prospects of NEETs: Evidence from Catalonia (Spain). Educational Research 57: 151–167.
Farrugia, D. (2018) Spaces of youth work, citizenship and culture in a global context: Youth, young adulthood and society. Routledge.
Thanks for this suggestion. We have introduced these two references when we talk about the importance of the education system as suggested.
I liked the international comparison of educational systems and youth transitions across European countries. I suggest the authors also refer to intra-national variation of educational systems, youth employment patterns, intervention programs, youth transitions and youth inactivity across the European regions. Literature on this issues has been recently increasing and highlighting this would place the paper better in the wider academic debates, thus increasing its value and contribution. Illustrative works that could be used include the following:
Avagianou, A., Kapitsinis, N., Papageorgiou, I., Strand, A.H., and Gialis, S. (2022) Being NEET in Youthspaces in the EU South. A Post-recession regional perspective. YOUNG 30, 425-454.
Kapitsinis, N., Poulimas, M., Emmanouil, E. and Gialis, S. (2022) Spatialities of being NEET in an era of turbulence: a regional resilience perspective for the Mediterranean EU South. Journal of Youth Studies https://doi.org/10.1080/13676261.2022.2101355
Thanks again. We have introduced these two references and have even introduced the concept of the relevant intra-national variation in the educational systems and other labour market indicators. We reported even examples of policies addressed only to specific regions due to the strong socio-economic differences within the same country. This is for example the case of Italy, that is one of the countries selected for the analysis in this article.
The section ‘Theoretical Framework’ should be renamed. It does not present theories or concepts, rather than the empirical background of the case study countries vis-à-vis the European context. Therefore, I suggest renaming it to ‘Empirical context’
Thanks for this observation. The the name of the section titled “Theoretical framework” has been changed into “Empirical context” as required.
The main weakness of the paper is the lack of a conceptual framework (which is a necessary ingredient for an academic paper) introducing the main concepts and theories about the education system and school to work transition. The authors should write a section between the introduction and the current second section which is called ‘Theoretical Framework’ but does not present theories or concepts, apart from the first paragraph.
We introduced a new section as required between the introduction and the “Empirical context” section describing the theoretical framework.
The ideas and arguments of the conceptual framework should be linked to the findings in the section of the analysis, by adding an extra section between the analysis and conclusion that will discuss the results in depth, connect them with the conceptual framework and then highlight their importance for the wider academic debates on education systems and their international variation. In other words, the paper lacks engagement with relevant literature (analysis, discussion and conclusion). This should be strongly revised and significantly expand the analytical character of the manuscript, which at the moment is rather descriptive.
An additional part was added to the manuscript discussing the results, before making the conclusions.
Additionally, the authors should add a brief section where they present and explain the methodological steps they followed to conduct this research and write this paper.
We inserted the methodology section were we described the steps followed to conduct this research
Finally, a basic issue that the authors should provide their own explanation in the conclusion, based on the results, from a critical point of view is what the national government should do about the relationship between education and labour market: should the education system be subordinated to the labour market needs, should the education system be more connected with the labour market needs or should the education system prepare students to fulfil the real needs of the society beyond the business interests? I think this question about education system vis-à-vis labour market is key when presenting and explaining studies related to contemporary education.
Thanks for this interesting comment that gave us the opportunity to reflect on the primary role of the education system and better develop the concept. In the conclusion, it has been clearly exposed.
Comments on the Quality of English Language
Some minor editing of English language would improve the paper
The authors have corrected some mistakes and grammar errors in the paper.
We hope that now its level of English could be considered good.
Reviewer 3 Report
Overall, the paper addresses an important problem, but there are areas that could benefit from improvement.
Firstly, it would be beneficial to include a dedicated theoretical perspective section. This section could explore different theoretical perspectives related to the topic, and it should be supported with appropriate references. For instance authors allocated only one paragraph without using any reference.
Secondly, the research question should be more explicitly stated. Clearly defining what the authors are looking to investigate and their motivation behind the study would enhance the paper's clarity and focus.
Regarding case selection, additional justification would be helpful, especially in terms of the similarity or difference between the selected countries. Considering welfare regimes could be one approach to strengthen the case selection rationale, particularly when dealing with countries like Latvia and Moldova from the same welfare regime families.
Presenting the findings in a more coherent and comparable way is essential for a comprehensive analysis. Creating a table to present the comparative findings could enhance the paper's organization and help readers to better understand the relationships between the different cases.
Lastly, the conclusion should be closely tied to the preceding discussions. It should provide a clear summary of the main findings and how they relate to the research question and theoretical framework.
In conclusion, addressing these areas of improvement could significantly enhance the paper's overall quality and impact. The authors have the opportunity to strengthen their research by considering these suggestions and providing more clarity and coherence throughout the paper.
Author Response
Overall, the paper addresses an important problem, but there are areas that could benefit from improvement.
Firstly, it would be beneficial to include a dedicated theoretical perspective section. This section could explore different theoretical perspectives related to the topic, and it should be supported with appropriate references. For instance authors allocated only one paragraph without using any reference.
Thanks for this valuable suggestion. We developed a new section for the theoretical framework and introduced many references to the literature on the topic
Secondly, the research question should be more explicitly stated. Clearly defining what the authors are looking to investigate and their motivation behind the study would enhance the paper's clarity and focus.
In the introduction we have now better exposed the motivation under our study
Regarding case selection, additional justification would be helpful, especially in terms of the similarity or difference between the selected countries. Considering welfare regimes could be one approach to strengthen the case selection rationale, particularly when dealing with countries like Latvia and Moldova from the same welfare regime families.
In the new section addressing the theoretical framework, we start with the description of the transition regimes and then clarify the relevance of studying the education system characteristics and reforms for both EU and non-EU countries.
Presenting the findings in a more coherent and comparable way is essential for a comprehensive analysis. Creating a table to present the comparative findings could enhance the paper's organization and help readers to better understand the relationships between the different cases.
We used table 1 to compare the situation of the education systems and labour markets in the countries analysed. Then we describe some reforms of the education system implemented in each country in reason of the specific issue of each country. These intervention programmes are not directly comparable. For this reason, in our opinion, the creation of a table to present comparative findings is not very useful
Lastly, the conclusion should be closely tied to the preceding discussions. It should provide a clear summary of the main findings and how they relate to the research question and theoretical framework.
In conclusion, addressing these areas of improvement could significantly enhance the paper's overall quality and impact. The authors have the opportunity to strengthen their research by considering these suggestions and providing more clarity and coherence throughout the paper.
Thanks for this observation. We have now connected more directly the conclusions with the rest of the paper and even improved them giving some qualitative reflection on the role of the education system that the current and future reforms should not forget. In particular, to better connect the conclusions with the rest of the paper, we included the same concepts presented in the first section of the conclusions in the introduction (highlighted in red, referred to the changes in he labour market and needs and the Bologna process).
Reviewer 4 Report
The study's purpose, theoretical framework, background information, and conclusions are well-presented and enable readers to grasp the challenges and initiatives being implemented. To further improve the paper, I would suggest considering the following points:
(i) Elaborate on employer perspectives: As the effectiveness of school-to-work transitions largely depends on the effective alignment of learning goals with actual industry requirements, it would be beneficial to understand the challenges and needs faced by organizations in terms of needed skills.
(ii) Update information and statistics: “Table 1. Statistics data comparing Italy, Moldova, Latvia in 2018” is a good reference but is five years old. Ensuring the paper reflects the most up-to-date information will reinforce the paper's findings and comparisons.
(iii) Expand on limitations and future research: Acknowledge any limitations of the study and propose avenues for future research, which can help foster further exploration and advancement in the field. Emphasize the importance of further exploring the actions taken by each country in fostering successful partnerships and collaborations among the government, employers, trade unions, and educational institutions to enhance school-to-work transitions.
Overall, the paper provides valuable insights into national support systems for school-to-work transitions, but incorporating the above suggestions can strengthen its impact and provide a more comprehensive understanding of the topic.
Author Response
The study's purpose, theoretical framework, background information, and conclusions are well-presented and enable readers to grasp the challenges and initiatives being implemented. To further improve the paper, I would suggest considering the following points:
- Elaborate on employer perspectives: As the effectiveness of school-to-work transitions largely depends on the effective alignment of learning goals with actual industry requirements, it would be beneficial to understand the challenges and needs faced by organizations in terms of needed skills.
Thanks for this useful suggestion. We have now better developed and described the connection between the education system and the labour market. In the introduction, for example, we talk about the importance of preparing young people to the new labour market requirements, in terms of skills and new paradigms, related to overcoming the concept of permanent work.
- Update information and statistics: “Table 1. Statistics data comparing Italy, Moldova, Latvia in 2018” is a good reference but is five years old. Ensuring the paper reflects the most up-to-date information will reinforce the paper's findings and comparisons.
Data in table 1 are updated to the most recent ones, that for the majority of the indicators is 2022.
- Expand on limitations and future research: Acknowledge any limitations of the study and propose avenues for future research, which can help foster further exploration and advancement in the field. Emphasize the importance of further exploring the actions taken by each country in fostering successful partnerships and collaborations among the government, employers, trade unions, and educational institutions to enhance school-to-work transitions.
We have now introduced a new section, before the conclusions, describing limits of the present research and future possible developments
Overall, the paper provides valuable insights into national support systems for school-to-work transitions, but incorporating the above suggestions can strengthen its impact and provide a more comprehensive understanding of the topic.
Thanks for appreciating our work. We hope to have fulfilled all the suggestions in the proper way.
Round 2
Reviewer 1 Report
My main concern after this review is that maybe the contribution of the paper should be explicit clearer in discussion or conclusion. Congratulations! The paper has improve tremendously.
Best regards.
Reviewer
.
Reviewer 2 Report
I would like to thank the authors for addressing the comments.
I think a deep and thorough editing of the English language would significantly improve the paper
Reviewer 3 Report
The paper has been significantly improved after revisions.
Reviewer 4 Report
Thanks for reviewing and incorporating the changes.
